# Masticatory Behaviors and Gender Differences in People with Obesity as Measured via an Earphone-Style Light-Sensor-Based Mastication Meter

**DOI:** 10.3390/nu14142990

**Published:** 2022-07-21

**Authors:** Nagisa Hidaka, Satoshi Kurose, Nana Takao, Takumi Miyauchi, Sachiko Nakajima, Sawako Yoshiuchi, Aya Fujii, Kazuhisa Takahashi, Hiromi Tsutsumi, Daiki Habu, Kazuhiro Taniguchi, Yutaka Kimura

**Affiliations:** 1Department of Health Science, Kansai Medical University, Hirakata 573-1010, Osaka, Japan; kurosesa@hirakata.kmu.ac.jp (S.K.); tsutsumi@makino.kmu.ac.jp (H.T.); kimuray@hirakata.kmu.ac.jp (Y.K.); 2Faculty of International Studies, Osaka Sangyo University, Daito 574-8530, Osaka, Japan; 3Health Science Center, Kansai Medical University Hospital, Hirakata 573-1010, Osaka, Japan; takaonan@hirakata.kmu.ac.jp (N.T.); miyaucht@hirakata.kmu.ac.jp (T.M.); sachiko-nakajima@fol.hi-ho.ne.jp (S.N.); tashimas@hirakata.kmu.ac.jp (S.Y.); ayafujii1012@gmail.com (A.F.); takakazu@hirakata.kmu.ac.jp (K.T.); 4Graduate School of Human Life and Ecology, Division of Human Life and Ecology, Osaka Metropolitan University, Osaka-shi 558-8585, Osaka, Japan; habu@life.osaka-cu.ac.jp; 5Faculty of Human Ecology, Department of Aesthetic Design and Technology, Yasuda Women’s University, Hiroshima-shi 731-0153, Hiroshima, Japan; taniguchi-k@yasuda-u.ac.jp

**Keywords:** obesity, masticatory behaviors, earphone-style light-sensor-based mastication meter, gender differences in masticatory behaviors

## Abstract

While people with obesity have been found to chew fewer times and for shorter durations, few studies have quantitatively evaluated mastication among this group. This study examined the relationship between the mastication characteristics of people with obesity and the factors correlated with obesity. To this end, 46 people with obesity and 41 healthy participants placed an earphone-style light sensor in the aperture of their outer ear. We also examined the partial correlation between this, their body composition, and various biochemical markers by gender. A two-way analysis of variance (ANOVA) regarding the masticatory index, gender, and the presence/absence of obesity for all three food items revealed the main effects in the gender difference and the presence/absence of obesity. Additionally, the number of times the salad was chewed showed an interaction between the gender and the presence/absence of obesity. In the BMI-corrected partial correlation analysis of the chewing index and the glucose/lipid metabolism index, the chewing time and the number of chews of all the food items negatively correlated with hemoglobin A1c(HbA1c), fasting plasma glucose (FPG), immunoreactive insulin (IRI), and homeostasis model assessment of insulin resistance (HOMA-R) in the female obese group. These findings might be used in weight-loss interventions for men with obesity and treatments that target the metabolic function among women with obesity.

## 1. Introduction

Obesity is a risk factor for many lifestyle diseases, including hypertension, diabetes, and heart attacks [1], with its treatment and prevention being important for extending peoples’ overall life expectancy. Obesity is caused by various factors; being a “fast eater” (that is, eating food within a relatively short period of time and masticating fewer times) is thought to be one. In practice, the notion that fast eaters are more prone to obesity has been reported in numerous studies. For example, one study demonstrated that undergraduate students who are fast eaters are 4.4 times more likely to become obese compared to slower-eating students, and that the risk of obesity in men is 2.8 times than that in women [2]. Another study, in which electromyogram readings were used to observe masticatory behaviors and compare participants’ chewing of one bite of food for either 35 or 10 times, found that chewing 35 times results in a faster chewing tempo and doubles the amount of time required to feel full, thereby decreasing the overall food intake [3]. A third study compared the diet-induced thermogenesis (DIT) in adult participants three hours after eating 621 kcal worth of food either quickly or slowly and found that their postprandial energy expenditure was 15 kcal in the former scenario while being 30 kcal in the latter (i.e., double the amount) [4]. There is thus a clear connection between a person’s number of chews and mastication time and their overall energy expenditure and satiety and, as such, dietary counseling and plans that incorporate these factors are more likely to be effective.

Studies have predominantly relied on participants’ self-assessment in terms of them being “fast” or “slow” eaters, although both of these terms are subject to individual interpretation [5] and cannot adequately capture masticatory behaviors. Other studies have used observational methods, including video footage [6]; however, because counting each participant’s number of chews requires a large quantity of data and, thus, a significant amount of time and work from the researchers, these methods have several limitations when it comes to large-scale studies. Alternatively, other methods, including measurements taken via electromyograms or sensors worn on the face, are viable, although they also involve a great deal of work.

Recent technological advancements have spurred the development of chewing meters that use the movement of the external ear canal [7,8], with user-friendly instruments now being available. While most previous studies have focused on overweight and healthy adults, no studies have yet examined the masticatory behaviors of people with obesity. Furthermore, no study has verified the relationship between masticatory behavior and the glucose/lipid metabolism index, as well as the relationship between masticatory characteristics, gender differences, and the blood index for each food item by objectively measuring Japanese people with obesity. Doing so might help advance the treatment of obesity. Therefore, in this study, we examined the relationship between the mastication habits of Japanese people with obesity and the factors correlated with the obesity blood index, in relation to glucose metabolism and insulin resistance, by using a light-sensor-based mastication meter. This mastication meter helped measure the number of chews and mastication time via the external auditory canal.

## 2. Materials and Methods

### 2.1. Participants

A group of 46 people with obesity who were outpatients at Kansai Medical University Hospital and whose body mass index (BMI) was 30 or higher (age 40.8 ± 11.3 years, BMI 39.3 ± 6.8), as well as a healthy group of 41 volunteers (age 39.4 ± 13.4 years, BMI 20.6 ± 1.9), participated in this study (Table 1).

The inclusion criteria for the obese group included those who were people aged 20 years or older, were independent when their consent was obtained, were receiving lifestyle disease counseling at Kansai Medical University Hospital Health Science Center, who understood the device used in the study, who could undergo the examinations in the study, and who voluntarily signed a consent form after receiving a thorough explanation of what participation in this study would entail. Inclusion criteria within the healthy volunteer group included those who were 20 years of age or older, were independent when their consent was obtained, understood the device used in the study, could undergo the examinations in the study, were staff or authorized personnel at Kansai Medical University Hospital Health Science Center, and who voluntarily signed a consent form after receiving a thorough explanation of what participation in this study would entail.

Exclusion criteria included those with chewing disorders or oral diseases due to cranial neuropathy, those with inflammation of the outer ear (otitis externa), those with any mental illness(es), those for whom participating in this study was determined to be difficult, and all others who were deemed inappropriate as research participants by the principal investigator of the study.

This study was conducted with the approval of the Kansai Medical University Institutional Review Board (approval #1647) and the Osaka Sangyo University Institutional Review Board (application #2016-ethics-019). All the procedures performed in the study involving human participants were performed in accordance with the revised Helsinki Declaration and its later amendments. Written informed consent was obtained from the patient before undergoing all clinical procedures.

### 2.2. Body Composition

Body composition was measured using the InBody 720 (BIOSPACE, Seoul, South Korea) while participants were in a fasted state. The items evaluated were participants’ weight, body fat, and BMI. BMI was calculated using the formula $\mathrm{weight}\;\mathrm{in}\;\mathrm{kg}/{\left(\mathrm{height}\;\mathrm{in}\;m\right)}^{2}$. Additionally, measurements of participants’ visceral and subcutaneous fat areas in the cross-section above their navel were taken using computed tomography (CT; GE Healthcare). CT and fat-scan analysis software (East Japan Technology Tokyo Laboratory, Tokyo, Japan) were used to measure the umbilical-level visceral fat area (VFA) and the subcutaneous fat area, respectively.

### 2.3. Biochemical Examination

Blood samples were taken in the early morning while participants were in a fasted state. The biochemical markers used for this study included fasting plasma glucose (FPG), hemoglobin A1c (HbA1c_NGSP), triglycerides (TG), low-density lipoprotein (LDL), high-density lipoprotein (HDL), γ-Glutamyl TransPeptidase (γ-GTP), immunoreactive insulin (IRI), and homeostasis model assessment of insulin resistance (HOMA-R). HOMA-R was used as a marker of insulin resistance and was calculated as insulin resistance index=fasting insulin×fasting glucose/405.

### 2.4. Evaluation of Mastication

The device used was the “earable” by eRCC. Participants placed the earphone-style light sensor (“earable” mastication meter, eRCC) in the aperture of their outer ear. Its earphones were then used to measure the number of chews and mastication duration via infrared rays and the optical distance sensors (LEDs and phototransistors) housed within the device to measure minute changes in the ear canal. In addition to showing the measured changes as a waveform, the device uses a proprietary algorithm to identify mastication and displays the number of chews in real time on a tablet. Because using the device is simply a matter of putting on a pair of earphones, it is easier to measure mastication indices under conditions that are relatively close to those experienced in everyday life (Figure 1) than it is with other devices that have been used previously that are worn around the jaw.

A goodness-of-fit index of 0.958 or greater, as well as a recall index of 0.937 or greater, have both been reported for this device, indicating that there is a high likelihood that it accurately counts the number of chews [8].

### 2.5. Foods

The designated foods in this study were:Salads (39 kcal, 1.2 g protein, 0.5 g fat, 6.2 g carbohydrate, and 0.2 g salt equivalent), which contained julienned cabbage and fiber-rich foods;Rice balls (188 kcal, 5.6 g protein, 1.6 g fat, 37.8 g carbohydrates, and 0.9 g salt equivalent);Donuts (214 kcal, 3.0 g protein, 10.4 g fat, 27.4 g carbohydrates (26.9 g sugar, 0.5 g dietary fiber), and 0.5 g salt equivalent).

Rice balls (*onigiri*) were utilized because they are a staple food that is liked by most Japanese people, with donuts being chosen because they are an easy-to-eat snack of about the same size as a rice ball.

### 2.6. Statistical Analyses

The participants’ chewing duration and number of chews were measured for each designated food (salad, rice ball, and donut), with the two groups’ mastication indices then being compared. In the obese group, the relationships between the mastication indices and participants’ biochemical markers were also examined.

The measured indices were represented as mean ± standard deviation. Normality was tested using a Shapiro–Wilk test. The number of chews and the chewing duration by gender for each designated food item in both groups were compared by performing a two-way analysis of variance. If a significant difference was observed, the Bonferroni method was applied to perform multiple comparisons. In addition, the biochemical markers that were measured in the obese group were examined by performing a BMI-corrected partial correlation analysis. SPSS ver. 25 was used as the statistical processing program.

## 3. Results

The participants’ attributes are shown in Table 1. In the group of people with obesity, the body weight and BMI of men were significantly higher than those of women, but no significant difference in age was observed. The number of chews and their overall duration for each food by gender is shown in Table 2. The mastication index of the three food items demonstrated the main effect by gender. Men had a significantly shorter mastication time than women, and their number of mastications was significantly less. In addition, the main effect was observed in the presence/absence of obesity. The chewing time for rice balls was significantly shorter in the obese group than in the healthy groups, and the number of times the three food items were chewed was significantly less. The number of times the salad was chewed was observed as an interaction. The chewing time for each of the three items was significantly longer in the order of salad, rice balls, and donuts, regardless of gender or obesity, and the number of times the items were chewed was the same.

Furthermore, no correlation was found in the partial correlation analysis between the index and the blood index of men in the obese group.

However, the partial correlation analysis of women demonstrated that their salad mastication duration was significantly negatively correlated with their HbA1c and FPG. Additionally, their rice ball chewing duration was found to be significantly negatively correlated with their FPG, IRI, and HOM-R. However, their mastication duration was found to be significantly positively correlated with their LDL. Finally, their donut chewing time was significantly negatively correlated with age, HbA1c, FPG, IRI, and HOMA-R, with the number of chews for donuts being significantly negatively correlated with their age and HbA1c (Figure 2).

## 4. Discussion

This study observed that mastication time and frequency differ depending on gender and the presence/absence of obesity. Among three food items, the number of times the salad was chewed showed an interaction between gender and the presence/absence of obesity.

We believe that these results, as based on an objective evaluation of mastication indices using an earphone-style light sensor, support the prevailing hypothesis that fast-eating is related to obesity [2,3,4]. Typically, after a person eats and their glucose level rises, this stimulates their satiety center and they feel full, causing them to stop eating. However, due to the fact that the satiety center is not immediately stimulated when a person’s glucose level rises, it is important for people to allocate enough time for the sensation of being full to set in [9,10]. In the case of fast eating, the possibility of overeating and excessive energy intake is high due to insufficient stimulation of the satiety center in terms of time. In addition to stimulating the satiety center so that it induces the feeling of fullness, histamines are also known to be involved in the breaking down of visceral fat and in the suppression of lipid synthesis [11,12]. When the number of chews is low, it has been found that there is an insufficient internal secretion of histamines, resulting in fat accumulation and a person then not feeling full after eating, which then causes an excess energy intake [13]. This then shows that, because of these biological mechanisms, shorter chewing durations and a relatively low number of chews are both causal factors in obesity, thereby supporting the results of this study.

In addition, the fact that the chewing duration and number of chews in this study were shorter and fewer for men with obesity than for women with obesity suggests that there are gender differences in masticatory behaviors. Gender differences in mastication have been reported before; for example, according to the research of Soojin Park et al. (2015), men were found to take in a greater amount of food per bite, their masticatory force was found to be greater, and their rate of food intake was higher than women, while women were found to exhibit a greater number of chews and to spend more time eating than men [14]. This suggests that masticatory guidance programs for people with obesity would be more effective if gender differences were taken into account.

Obese or not, men have been found to be inherently fast eaters [15] and to exhibit different masticatory behaviors compared to women. It has also been suggested that this fast-eating tendency intensifies as one’s degree of obesity increases; as such, masticatory guidance interventions would also be more effective for overweight men whose obesity has not yet reached an advanced stage (e.g., prior to them reaching a BMI of 35). Going forward, future research will need to be performed on the factors that accelerate masticatory behaviors in men, regardless of whether they are obese or not.

The interaction between gender and obesity was observed in the number of chews for salad, because salad contains a lot of fiber and requires more chewing than other food items, thereby demonstrating a distinct difference in chewing behavior. In other words, the chewing behavior of people with obesity, considering gender, may be clarified by using fibrous food items, such as salad, rather than carbohydrate-rich food items such as rice balls and donuts.

Among the participating women, the only significant difference found was in the masticatory behaviors of the obese group regarding the included staple food (i.e., the rice balls). In addition, their masticatory behaviors for other foods were found to be negatively correlated with their body composition and metabolic markers. It has been previously reported that women exhibit a greater number of chews and longer chewing durations than men [14]. Because men exhibit a greater bite force than women, this is a possible reason for why women need to chew more and for longer. The fact that a significant difference was only found regarding rice balls may be due to the fact that they are made up of individual grains, have a higher water content than the other included foods, and are easier to chew than salad or donuts. Meanwhile, salad is high in fiber and donuts are more solid; as such, women would need to chew them relatively more times due to their weaker bite force, which suggests that a difference in the masticatory behaviors between obese and healthy women would be less likely.

The fact that the participating women’s masticatory behaviors were correlated with more of their biomarkers when compared to the men suggests that effective masticatory guidance for women may contribute to weight loss and improved metabolic function by partial correlation analysis. In particular, the masticatory indices for high-carbohydrate foods, including rice balls and donuts, are correlated with blood glucose and insulin resistance, which suggests that rectifying one’s fast-eating habits could positively affect their overall glucose metabolism. As such, it may be more effective to focus primarily on foods made up of carbohydrates and grains in providing masticatory guidance to women. Furthermore, a positive correlation was observed between the chewing time for rice balls and LDL cholesterol among obese women, but the detailed mechanism is unknown. In this regard, the masticatory index might have been related to glucose metabolism rather than lipid metabolism in women.

Although there are many reports on masticatory behavior and body shape in previous studies [2], this study is the first to evaluate Japanese people with obesity using an earphone-type sensor. The method of attaching an earphone-type sensor to the ear to evaluate the chewing time and frequency is easy and non-invasive and might be beneficial to the treatment of obesity.

This study does have several limitations. First, due to its cross-sectional nature, any causal relationships between participants’ masticatory behaviors for each food and their biochemical markers were not able to be identified. Second, while the study did find gender differences in masticatory behaviors, it did not sufficiently examine the relationship between these behaviors and sex-related hormones, which are thought to be behind those differences. Going forward, longitudinal studies will need to be conducted on the weight-loss effects of masticatory guidance programs or of changes in masticatory behaviors and their resulting correlations with biochemical markers.

## 5. Conclusions

While the duration of mastication and overall number of chews varied between the designated food types in this study for the obese and healthy groups, the former was found to exhibit shorter durations of mastication and a lower number of chews overall. Gender differences in masticatory behaviors were also observed, particularly with regard to the correlation between the participating obese women’s masticatory behaviors and several of their biomarkers. These findings suggest that providing masticatory guidance to obese patients might help them in achieving weight loss and improved metabolic function. A longitudinal interventional study will be needed to verify this.

## Figures and Tables

**Figure 1 nutrients-14-02990-f001:**
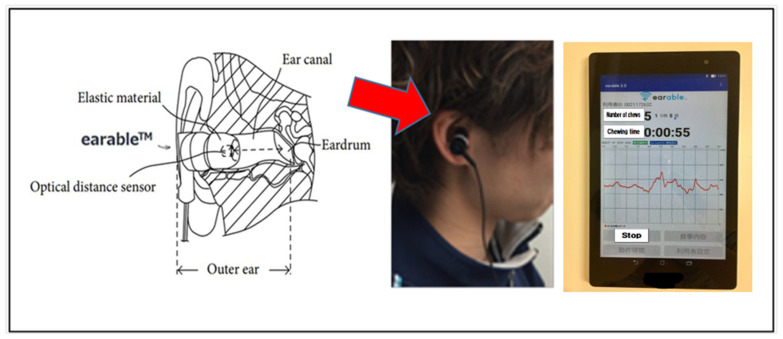
Structure of the earable, a diagram of it in use, and what it displays on a tablet.

**Figure 2 nutrients-14-02990-f002:**
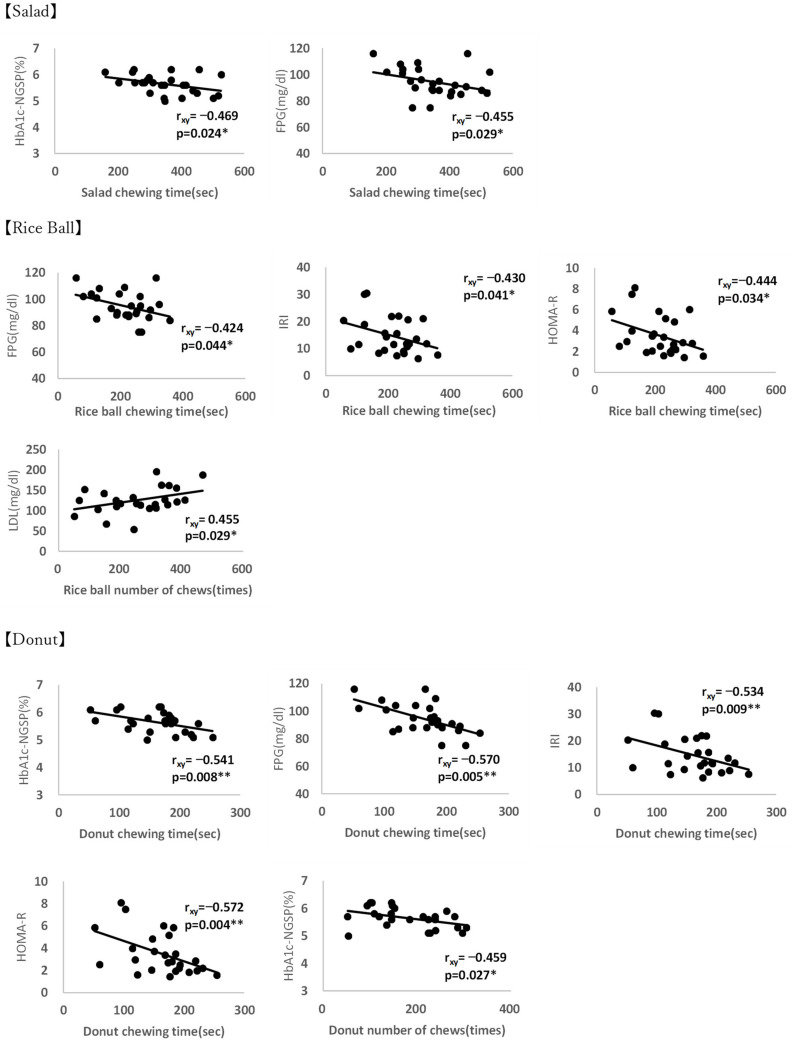
Partial correlation analyses of the duration of mastication and number of chews with each biomarker among women with obesity. *: means *p* < 0.05, **: means *p* < 0.001.

**Table 1 nutrients-14-02990-t001:** Participant attributes.

Index	Men			Women		
	Obese Group (*n* = 20)	Healthy Group (*n* = 21)	*p*-Value	Obese Group (*n* = 26)	Healthy Group (*n* = 20)	*p*-Value
Age (years)	42.5 ± 13.9	41.3 ± 14.2	0.803	39.5 ± 9.0	37.3 ± 12.6	0.484
Height (cm)	171.3 ± 6.1	172.7 ± 7.0	0.543	158.8 ± 6.5	159.1 ± 4.8	0.793
Weight (kg)	117.9 ± 17.2	64.5 ± 8.2	<0.001 **	95.4 ± 14.2	49.6 ± 4.0	<0.001 **
Body fat (%)	38.5 ± 4.1	N/A	N/A	50.2 ± 4.4	N/A	N/A
BMI (kg/m^2^) ^†^	39.6 (35.8–44.9)	21.6 (20.1–23.0)	<0.001 **	36.1 (34.340.7)	19.6 (18.3–21.1)	<0.001 **
Subcutaneous fat (cm^3^)	521.4 ± 156.6	N/A	N/A	498.3 ± 157.7	N/A	N/A
Visceral fat (cm^3^)	235.9 ± 96.8	N/A	N/A	166.6 ± 59.5	N/A	N/A

^†^: A Mann–Whitney test was to analyze this variable. Data are shown as mean ± standard deviation or median (25–75% range), BMI = body mass index. **: means *p* < 0.001.

**Table 2 nutrients-14-02990-t002:** Comparison between the study groups’ mastication duration and number of chews for each food.

Index	Men		Women		*p*-Value		
Chewing Duration	Obese Group (*n* = 20)	Healthy Group (*n* = 21)	Obese Group (*n* = 26)	Healthy Group (*n* = 20)	Sex Effect	Effect of Obesity	Sex × Obesity Interaction
Salad (s)	228.5 ± 106.6	282.7 ± 93.1	350.2 ± 95.9	348.4 ± 85.2	<0.001	0.210	0.180
Rice ball (s)	130.7 ± 56.9	181.0 ± 55.8	216.5 ± 77.3	266.5 ± 60.7	<0.001	0.001	0.989
Donut (s)	123.4 ± 39.8	137.9 ± 42.4	162.5 ± 50.3	187.2 ± 60.4	<0.001	0.067	0.635
Number of chews							
Salad (times)	237.1 ± 106.5	386.0 ± 136.7	397.8 ± 150.6	431.7 ± 120.7	0.001	0.002	0.047
Rice ball (times)	138.2 ± 65.4	257.2 ± 103.6	254.7 ± 113.0	327.9 ± 101.6	<0.001	<0.001	0.285
Donut (times)	119.2 ± 36.8	171.3 ± 73.3	181.7 ± 75.8	216.6 ± 72.8	<0.001	0.004	0.556

Data are shown as mean ± standard deviation. Two-way ANOVA was used.

## Data Availability

Data sharing is not applicable to this article.

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
