# Peer review of "Masticatory Behaviors and Gender Differences in People with Obesity as Measured via an Earphone-Style Light-Sensor-Based Mastication Meter"

_nutrients, 2022, doi:10.3390/nu14142990_

Round 1

Reviewer 1 Report

Abstract:  Consider putting all the results for men together and then all the results for women together.  Bouncing back and for is a tad confusing.

Intro: The introduction was clear, organized and well written.

Methods: The first paragraph describing the number of subjects and their characteristics should be moved to the results section.  Using a CT scan as one measure of obesity is concerning.  Please provide more details about the scan (what body part(s)) and was this the only reason the subject had CT scans.   The figure showing the ear mastication measuring devise is very helpful to the reader.

Results:  Page 5, was the secondary analyses evaluating only subjects with "advanced" obesity planned, was it a priori? If not, this may not be appropriate to include especially given the smaller sample size.  If it was planned (before doing initial analyses) please add this to the methods section and be clear this was not an after thought.  This came across as being done because when all subjects were included the relationship was not significant.  

Discussion: Consider removing the first paragraph, it is not necessary.  On page 7, line 194, please do not use "cause".  While your study suggests that the rate and duration of chewing is associated with obesity, your data does not support it "causes" obesity.  Your data are different for men and women and depended somewhat on the type of food eaten.  While this is very interesting research more needs to be done to show a causal effect. Also, on page 8, line 252 you state a causal relationship is no possible.  

Page 9, line 266 please change "would help" to might/may help.  

Author Response

Dear Reviewer 1,

We apologize for the delay in replying to the revised manuscript.
We will attach you the answers to the peer-reviewed comments.

Reviewer 2 Report

Manuscript: ‘Masticatory behaviors and their gender differences in obese patients as measured via an earphone-style light-sensor-based mastication meter’ by Nagisa Hidaka; Satoshi Kurose; Nana Takao; Takumi Miyauchi; Sachiko Nakajima; Sawako Yoshiuchi; Aya Fujii; Kazuhisa Takahashi; Hiromi Tsutsumi; Daiki Habu; Kazuhiro Taniguchi; Yutaka Kimura

The authors describe outcomes of a cross-sectional study using a light-sensor-based mastication meter. The study entails comparison of obese and eutrophic participants eating distinct foods and measuring mastication indices. In addition, metabolic blood parameters are associated with mastication indices.

Major concerns

Statistic is incomplete in reporting (details of test and outcomes are missing) and the used statistic is not appropriate for the design. The authors investigate one continuous dependent variable and several categorical variables at the same time (gender, body type, and three food types) – this cannot be analyzed by using t-tests and U-tests. An independent factorial ANOVA is necessary to find main effects and interactions with appropriate validity. In addition, multiple repeated correlations used by the authors produce the danger of finding associations by chance – authors need to perform a Bonferroni’s correction for their outcomes. Some of the graphs with scatter plots look like that data are not complying with Pearson’s correlation test assumptions (i.e., homoscedasticity – Figure 2).

Multiple comparisons of parameters, in particular correlations of mastication indices with blood parameters, are without hypotheses and not prepared by background narrative in the introduction. Results seemed to be picked out due to significance but not based on theory and proper development of a hypothesis. Clearly, reported correlations between mastication parameters and blood parameters are driven by the association of BMI/fat with the blood parameters and collinearity with mastication indices. Without proper hypotheses and preparation of the idea of running these correlations by research background, the need for the analyses and the scientific justification are questionable.

The idea of the connection of satiation with mastication and connection with histamine is established on mice research; human research more point towards a different accessibility of nutrients and hormonal response with higher mastication in postprandial period (Hollis, 2018). However, measurements of satiety and postprandial blood parameters would be necessary if this type of hypothesis is stated; many parts in the discussion part exaggerate correlation findings and mix those with causality and findings not measured in the study.

Findings which are valid in the study context are potentially (if proper statistics is applied) the main effect of body type and gender, while both results are published already elsewhere.

Minor concerns

Use of terminology ‘healthy’ versus ‘obese’ is questionable – use of ‘eutrophic’ would be more appropriate.

Inflating results of outcomes with specific food types is poorly justifiable and main effect and interactions are not analyzed.

Many parts of the discussion read exaggerated in statements and interpretations speculative. Particular parts with focus on various food types and effects on satiation, blood parameters and mechanisms extrapolated from animal literature.

Effects of masticatory guidance programs are speculative and repeated several times in discussion.

Author Response

Dear Reviewer 2,

We apologize for the delay in replying to the revised manuscript.

We will attach you the answers to the peer-reviewed comments.

Round 2

Reviewer 2 Report

All concerns were reviewed and substantial improvements have been made to the manuscript.